# Mesenchymal stem cells promote metastasis through activation of an ABL-MMP9 signaling axis in lung cancer cells

**Jing Jin Gu[1], Jacob Hoj[1], Clay Rouse[2], Ann Marie Pendergast[1]***

**1** Department of Pharmacology and Cancer Biology, Duke University School of Medicine, Durham, North Carolina, United States of America, **2** Division of Laboratory Animal Resources, Duke University School of Medicine, Durham, North Carolina, United States of America

* ann.pendergast@duke.edu

**Data Availability Statement:** All relevant data are within the paper and its Supporting Information files.

## Abstract

Mesenchymal stem cells (MSCs) are recruited and activated by solid tumors and play a role in tumor progression and metastasis. Here we show that MSCs promote metastasis in a panel of non-small cell lung cancer (NSCLC) cells. MSCs elicit transcriptional alterations in lung cancer cells leading to increased expression of factors implicated in the epithelial-to-mesenchymal transition (EMT) and secreted proteins including matrix metalloproteinase-9 (MMP9). MSCs enhance secretion of enzymatically active MMP9 in a panel of lung adeno-carcinoma cells. High expression of *MMP9* is linked to low survival rates in lung adenocarci-noma patients. Notably, we found that ABL tyrosine kinases are activated in MSC-primed lung cancer cells and functional ABL kinases are required for MSC-induced MMP9 expres-sion, secretion and proteolytic activity. Importantly, ABL kinases are required for MSC-induced NSCLC metastasis. These data reveal an actionable target for inhibiting MSC-induced metastatic activity of lung adenocarcinoma cells through disruption of an ABL kinase-MMP9 signaling axis activated in MSC-primed lung cancer cells.

## Introduction

Lung cancer is the leading cause of cancer mortality worldwide with an overall five-year sur-vival rate of less than 20% [1, 2]. Notably, ~40% of lung cancer patients have metastasis at the time of diagnosis which is associated with increased morbidity and mortality [3]. Patients with non-small cell lung cancer (NSCLC) harboring mutations in the epidermal growth factor receptor (EGFR) tyrosine kinase are at high risk of developing metastasis following the emer-gence of therapy-resistance to EGFR targeted therapies [4, 5]. While much progress has been made on the identification of tumor-intrinsic oncogenic drivers and pathways implicated in lung tumor progression and metastasis, much less is known regarding the role of stromal cells in the regulation of lung cancer metastasis.

Mesenchymal stem cells (MSCs) are heterogenous stromal cells present in most tissues that play important roles in tissue regeneration, wound healing, as well as tumor progression and metastasis [6, 7]. MSCs have been shown to migrate to tumors, undergo activation, and engage

**Funding:** This work was supported by NIH NCI grants R01CA195549 (A.M.P.), F31CA22496001 (J.P.H.), F99CA245732-01 (J.P.H.), and 5T32GM007105-44 (J.P.H.), the Lung Cancer Research Foundation Free to Breathe Metastasis Research Grant (A.M.P.), the Emerson Collective, and the Duke SPORE in Brain Cancer grant (P50CA190991).

**Competing interests:** The authors have declared that no competing interests exist.

in bidirectional inter-cellular interactions with tumors and distal metastases [8]. Tumor-associated MSCs also interact with immune cells and other cell types in the tumor microenvironment and have been shown to induce resistance to chemotherapy [9, 10]. MSCs can inhibit growth of breast cancers and other tumors [11–13]. However, a large body of work supports tumor-promoting roles for MSCs at multiple stages of the metastatic cascade including invasion, epithelial-to-mesenchymal transition (EMT), pre-metastatic niche formation, and metastatic outgrowth [14–17].

MSCs induce pro-metastatic activities of diverse tumor types through activation of signaling networks mediated in part by cell surface receptors, secreted factors and metabolites [18]. Here we uncover a role for MSC-mediated ABL tyrosine kinase-dependent activation of matrix metalloproteinase-9 (MMP9) in lung cancer cells which is linked to enhanced metastasis. We previously showed that the ABL family of tyrosine kinases, ABL1 and ABL2, promote metastasis by breast and lung cancer cells in mouse models through activation of transcription factor networks [19, 20]. More recently, we uncovered a lung cancer cell-intrinsic feed-forward ABL2-TAZ-AXL signaling axis that promotes lung adenocarcinoma metastasis to the brain [21]. Here we report that ABL kinases are activated in lung cancer cells co-cultured with MSCs, and that ABL kinase activity is required for MMP9 secretion and activation in a panel of lung adenocarcinoma cells. These data reveal an actionable target for inhibiting MSC-induced metastatic activity of lung cancer cells through disruption of ABL kinase signaling leading to decreased production of MMP9 in lung cancer cells.

## Materials and methods

### Animal experiments

Mice were housed under pathogen-free conditions in the Duke University Cancer Center Isolation Facility. All mouse studies followed the protocols reviewed and approved by the Duke Institutional Animal Care and Use Committee (IACUC). Age matched athymic nu/nu mice (age 6–10 weeks old) (provided by Duke University animal breeding facility) were used for intracardiac injection for metastasis study. Tumor cells were labeled with pFU-Luciferase-Tomato (pFuLT) DNA to allow in vivo detection by bioluminescent imaging (BLI). Mice were anesthetized with 5% isoflurane prior to intracardiac injections or BLI imaging. Cancer cells suspended in 100 μL PBS were injected into the left cardiac ventricle with a 30-gauge needle. For BLI imaging, mice were injected intraperitoneally wit D-luciferin (150 mg/kg) followed by imaging using IVIS XR bioluminescent imager. All experimental mice were monitored until fully recovered from anesthesia and were subsequently monitored for disease progression and imaged for metastasis burden weekly. We monitored mouse behavior and health daily or more frequently depending on health condition once we began the experiment. Mice were monitored by measuring body weight and clinical signs (pain and distress) as described in our animal protocol. When we detected significant or accelerated losses in body weight (>15%), or mice under distress, we consulted with veterinary personnel and performed euthanasia as recommended. Mice were euthanized by $CO_2$ followed by secondary physical disruption such as exsanguination when they reached the humane endpoints criteria (approved in our animal protocol A062-19-03). No mice died before meeting criteria for euthanasia.

### Cell lines

Human NSCLC cell lines HCC827 (CRL-2868), HCC4006 (CRL-2871) and H1650 (CRL-5883) were purchased from ATCC. PC9 cells were a gift from Dr. Joan Massague (Memorial Sloan Kettering Cancer Center, New York, NY, USA) [22]. Cells were maintained in RPMI 1640 (Life Technologies) supplemented with 10% fetal bovine serum (FBS, Sigma), 10 mM

HEPES, 1 mM sodium pyruvate, and 0.2% glucose. Cell lines were grown and multiple vials were frozen after the first passage of the initial vial to ensure that only early passages of cell lines are used for all experiments. New vials were used monthly to reconstitute the working population. Fetal Bovine Serum (FBS) employed in cell culture media was extensively tested prior to purchasing the lots to ensure that the serum is endotoxin-free. HEK293T cells obtained from Duke Cell Culture facility were used for transfection and virus production and were maintained in DMEM (Life Technologies) with 10% FBS (Corning). All cell lines were tested routinely for mycoplasma (MycoAlert Plus, Lonza). Human bone marrow-derived mesenchymal stem cells (MSC, Lonza) were cultured in DMEM (low glucose, Life Technologies) supplemented with 10% FBS, 2 mM glutamine and antibiotics Pen/Strep. MSCs at passages 5–6 were used for all experiments. All cultures were maintained at 37°C in humidified incubator with 5% $CO_2$.

## Cell culture

NSCLC cells were co-cultured with MSCs at a 1:1 ratio. For all experiments, cells were either cultured in single culture or under co-culture conditions using 50% of MSC medium and 50% of NSCLC medium. Cells were cultured for 3 days, media was removed and cells were washed with PBS followed by culturing in Serum-free media for 24h. Culture supernatant was collected, centrifuged to remove debris and concentrated (Amicon Ultra 10K centrifugal filter, Millipore) for protein analysis. To separate NSCLC cells from MSCs after co-culture, MSC cells were labeled with Cell Tracker Violet (BMQC dye, Invitrogen) before mixing with NSCLC cells labeled with luciferase-tomato. Both cells in single cultures and co-cultures were FACS-sorted. For ABL kinase inhibition studies, we used the ABL-specific allosteric inhibitors GNF-5 and ABL001 (synthesized by the Duke University Small Molecule Synthesis Facility). Inhibitors were added to the culture and DMSO was used as solvent control.

## Real-time RT-PCR

RNA was isolated from cells using the illustra RNAspin Mini kit (GE Healthcare), and complementary cDNA was synthesized using oligo(dT) primers and Moloney murine leukemia virus reverse transcriptase (Invitrogen). Realtime PCR was performed using iTaq Universal SYBR Green Supermix (Bio-Rad). Primers used for this study are listed in S1 Table. Analysis was performed using a Bio-Rad CFX384 real-time machine and CFX Manager software. All qPCR assays were performed in triplicates. The expression of each gene was normalized to that of the GAPDH or 18S gene using the ddCT method (Bio-Rad).

## Viral transduction

HEK293T cells were transfected with lentiviral packaging cDNAs: pMDL, pCMV-VSVG, and pRSV-REV with either scrambled (SCR) or ABL1+ABL2-specific (AA) shRNAs [19] using FuGENE6 reagent (Promega). Culture supernatants containing lentiviruses were collected and filtered at 24 and 48 hours after transfection and used for transducing NSCLC cells in the presence of 8 μg/ml polybrene (Sigma-Aldrich). Cells transduced with lentiviruses encoding shRNAs were sorted for GFP+ cells by FACS.

## Immunoblotting

Cells were lysed in RIPA buffer (50 mM Tris-HCl, pH 7.5, 150 mM NaCl, 1% Triton X-100, 0.1% SDS, and 0.5% sodium deoxycholate with protease/phosphatase inhibitors). Cell debris was removed by microcentrifugation, and protein concentration was quantified using the DC Protein Assay (Bio-Rad Laboratories). Equal amounts of total cell lysates or concentrated

culture supernatant (normalized to total lysate protein) were separated by SDS–polyacryl-amide gel electrophoresis and transferred onto nitrocellulose membranes and probed with the indicated antibodies. Tubulin or Actin were used as loading controls. Antibodies used are listed in S2 Table. Protein gels were quantified using Fiji software.

## Zymography

Concentrated culture supernatant was analyzed on 7.5% SDS-polyacrylamide gel containing 0.1% gelatin (Sigma). Gels were washed 30 min twice in buffer containing 50 mM Tris (pH7.5), 5 mM $CaCl_2$, 2.5% Triton X-100 and 1mM $ZnCl_2$ followed by incubating in buffer containing 50mM Tris (pH7.5), 5mM $CaCl_2$, 1% Triton X-100 and 1mM $ZnCl_2$ for 24h in 37˚C. Gels were stained by Coomassie blue and destained in 40% Methanol and 10% acetic acid. The zymography gels were imaged (BioRad, ChemiDoc XRS) with Image Lab software and quantified using Fiji software.

## Angiogenesis array

PC9 cells transduced with lentiviruses carrying either scrambled (SCR) or ABL1/ABL2-specific (AA) shRNAs were cultured with or without MSCs for 72h followed by culture in serum-free media for another 24h. Culture supernatants were concentrated and analyzed using Human Angiogenesis Array (R&D Systems, Catalog #ARY007) following manufacturing instructions.

## Statistical analysis

Statistical analyses were performed using GraphPad Prism 8. Comparisons of two groups were performed using unpaired two-tail Student's t tests. Comparisons involving multiple groups were evaluated using one-way ANOVA, followed by Tukey's post hoc test. For all tests, $P < 0.05$ was considered statistically significant. For Kaplan-Meier survival analysis, p values were calculated using log-rank (Mantel-Cox) testing. Data shown represent averages ± SEM. Analysis of lung adenocarcinoma patient microarray data was performed using the KMplot analysis tool (kmplot.com) [23] using publicly available datasets and existing online analysis tools [23]. Patient groups were divided into tertiles (n = 719 total patients across all cohorts for OS, n = 461 total patients across all cohorts for PFS). The Affymetrix identifier for *MMP9* used in the analysis was 203936_s_at. Survival data were plotted in GraphPad Prism 8 software and statistical analysis was performed by Log-Rank Mantel-Cox testing.

## Results

### Mesenchymal stem cell (MSC)-primed lung cancer cells exhibit increased metastasis

In order to investigate the effect of MSCs on lung tumor progression and metastasis, we employed EGFR-mutant human PC9 and HCC827 NSCLC cells (labeled with a Tomato fluorescence reporter) that were primed by pre-incubation with or without bone marrow-derived MSCs for 3 days. PC9 and HCC827 lung cancer cells were also labeled with luciferase for detection by bioluminescence imaging (BLI). Equal numbers of cells primed with or without MSCs were used for intracardiac injections into nude mice (**Fig 1A–1C**). FACS analysis showed that MSCs comprise approximately 5–6% of the total number of cells co-cultured with Tomato+ lung cancer cells (**S1A Fig**). Interestingly, in vitro growth of PC9 lung cancer cells co-cultured (Co) with MSCs was slightly reduced compared to PC9 cells grown without MSCs under single (S) culture conditions (**S1B Fig**). Notably, mice injected with PC9 and HCC827 cells co-cultured with MSCs displayed a marked increase in metastasis (**Fig 1A–1C**) and

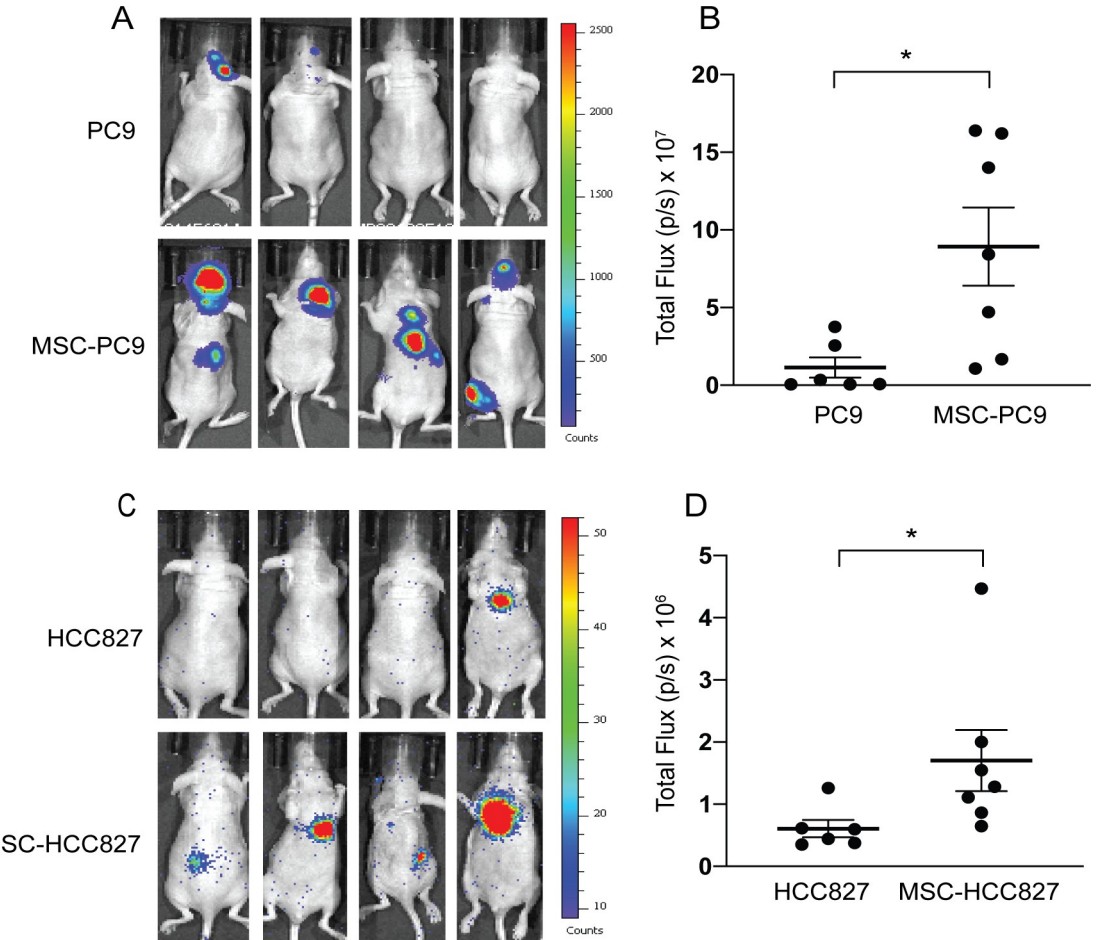

**Fig 1. Mesenchymal stem cells enhance metastasis of NSCLC cells.** PC9 and HCC827 cells were cultured with or without human bone-marrow derived mesenchymal stem cells (MSCs) (1:1 ratio) for 3 days. Equal numbers of cells were used for intracardiac injection ($4x10^5$ cells/mouse for PC9 and MSC-PC9; $1x10^5$/mouse for HCC827 and MSC-HCC827). Both PC9 and HCC827 cells were labeled with luciferase for detection by bioluminescence imaging (BLI). (A) Mice harboring PC9 and MSC-PC9 cells were imaged at day 28 post injection. Representative images are shown. (B) BLI total flux photons/sec (p/s) counts were plotted (PC9 n = 6; MSC-PC9 n = 7). (C) HCC827 and MSC-HCC827 mice were imaged at day 31 post injection. Representative images are shown. (D) BLI total flux (p/s) counts were plotted (HCC827 n = 6; MSC-HCC827 n = 7). Statistical analysis was performed using Student's unpaired two-tailed t test (* $p < 0.05$).

corresponding BLI counts (**Fig 1B–1D**). Similar results were observed with HCC4006 NSCLC cells harboring mutant EGFR, as co-culture with MSCs enhanced HCC4006 metastasis following intracardiac injection (**S2 Fig**). Together these data support a pro-tumorigenic role for MSCs leading to enhanced lung adenocarcinoma metastasis.

## MSCs induce EMT in lung cancer cells

MSCs have been reported to promote EMT in cancer cells [24]. To assess the functional effects of MSCs on NSCLC cells, NSCLC and MSC cells were differentially labeled with fluorescence markers and co-cultured, following which NSCLC cells were subjected to FACS sorting. The mRNA and protein expression of EMT markers Vimentin, Zeb1, Snail, and Slug were markedly increased in PC9 lung cancer cells co-cultured with MSCs compared to PC9 cells cultured in the absence of MSCs (**Fig 2A and 2B**). Similarly, expression of Vimentin and Snail proteins was increased in HCC827, HCC4006, and H1650 NSCLC cells upon co-culture with

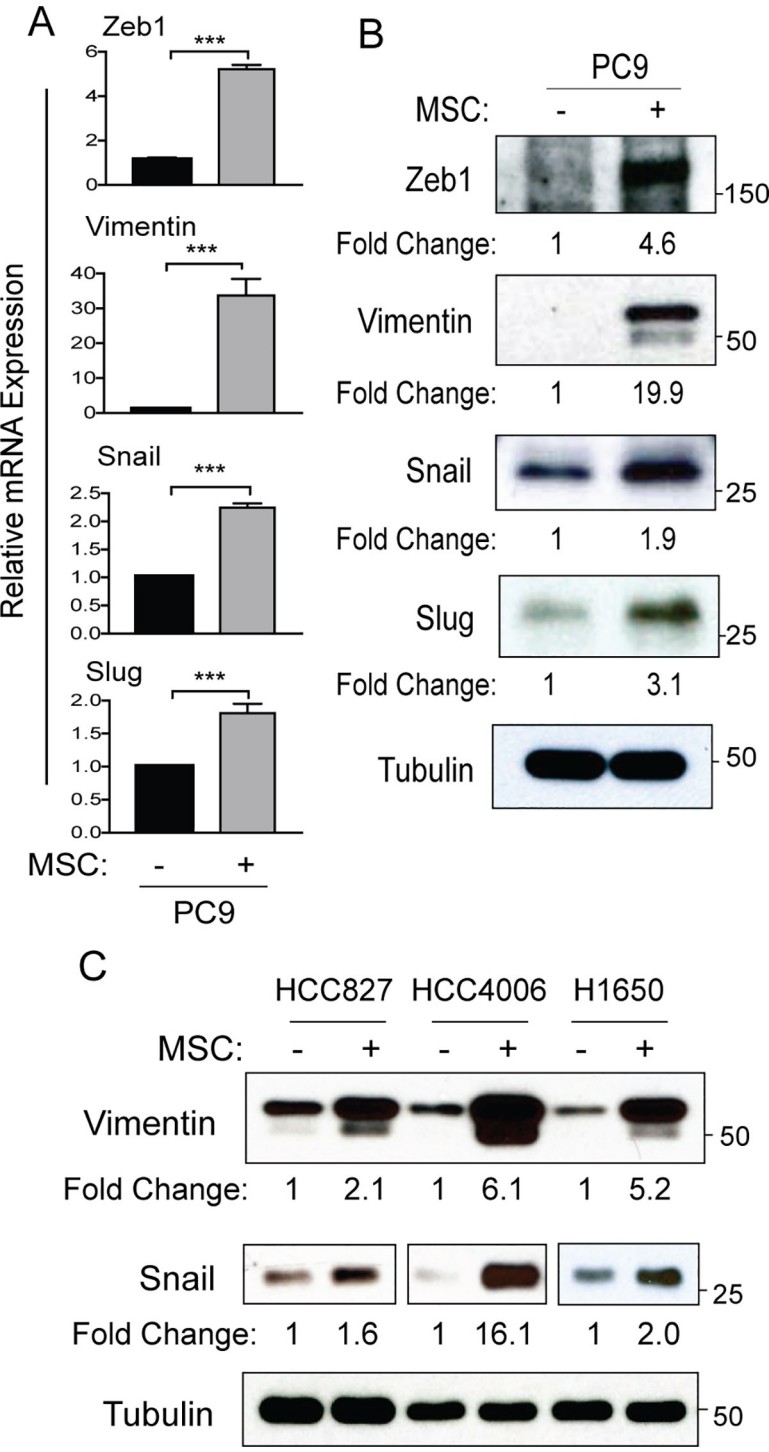

**Fig 2. MSCs induce EMT in lung cancer cells.** PC9 cells were cultured with or without MSCs followed by FACS sorting to separate PC9 (tomato+) from MSC (violet+) cells. (A) RT-PCR was performed for expression of indicated epithelial-mesenchymal transition (EMT) mRNAs in PC9 cells from single cell culture or sorted following co-culture with MSCs. Statistical analysis was performed using unpaired two-tailed t test (***p<0.001). Error bars represent ± SEM (n = 3). (B) Total lysates of PC9 cells from single culture and FACS-sorted PC9 cells following co-culture with MSCs were analyzed for expression of EMT proteins using indicated antibodies. (C) HCC827, HCC4006 and H1650 cells were cultured with or without MSCs followed by sorting. Total cell lysates from single cell culture and sorted from co-culture with MSCs were analyzed for expression of EMT proteins as indicated. Each protein was normalized to tubulin and presented as fold change shown under the protein blots.

MSCs (**Fig 2C**). MSC co-culture also increased mRNA expression of EMT markers in these lung cancer cells (**S3 Fig**). These findings are consistent with a role for MSCs in promoting EMT in various tumor types [25].

## MSCs induce enhanced expression and secretion of MMP9 in NSCLC cells

Interactions of tumor cells with distinct cell types in the tumor micro-environment (TME) result in permissive conditions for activation of the metastatic cascade [18, 26]. Among the proteins known to play essential roles in this process are matrix metalloproteinases (MMPs) which are a family of zinc-dependent endopeptidases that regulate extracellular matrix (ECM) remodeling and function to activate and release cytokines and growth factors required for cancer cell invasion and metastasis [27, 28]. Therefore, we evaluated whether expression of a panel of MMPs was altered in PC9 lung cancer cells and MSCs following co-culture and FACS sorting. Interestingly, we observed a profound induction of *MMP9* mRNA expression in lung cancer cells in response to co-culture with MSCs (**Fig 3A**, top panel). In contrast, mRNA expression of *MMP9*, which belongs to the gelatinase MMP subtype, was low in MSCs even under co-culture conditions. An MSC-induced increase in *MMP9* mRNA expression was also observed in HCC827, HCC4006, and H1650 NSCLC cells (**S4A Fig**). Transcriptional expression of *MMP7* (matrilysin) was significantly increased in MSCs and to a much lesser extent in PC9 and H1650 lung cancer cells following co-culture (**Fig 3A and S4B and S4C Fig**). Secretion of MMP9 protein and to a lesser degree MMP7 protein in the culture supernatant was increased following MSC co-culture and was inhibited by treatment with the ABL kinase inhibitor GNF5 (**S4D Fig**). Expression of *MMP1* (collagenase), *MMP2* (gelatinase), and *MMP3* (stromelysin) transcripts was not significantly altered in lung cancer cells co-cultured with MSCs, but all three transcripts were downregulated in MSCs after co-culture with lung cancer cells (**Fig 3A**). Expression of *MT1-MMP* (*MMP14*) was not significantly altered in MSCs and PC9 lung cancer cells cultured alone or under co-culture conditions (**Fig 3A**, bottom panel). Thus, among the MMP cohort analyzed, *MMP9* mRNA expression is selectively upregulated in lung cancer cells in the presence of MSCs.

To evaluate whether the profound increase in *MMP9* mRNA in lung cancer cells elicited by co-culture with MSCs resulted in increased MMP9 protein secretion, PC9 and HCC827 lung cancer cells were cultured with or without MSCs for 3 days, followed by serum starvation for an additional 24 hours. Analysis of secreted MMP9 in the culture supernatant demonstrated a large increase of MMP9 protein in the culture supernatant (SN) from PC9 and HCC827 lung cancer cells grown in the presence of MSCs compared to cancer cells grown without MSCs (**Fig 3B**). Similarly, MMP9 secretion was markedly enhanced in HCC4006 and H1650 lung cancer cells co-cultured with MSCs (**Fig 3C**). To assess the extent to which direct co-culture of lung cancer cells with MSCs affected MMP9 enzymatic activity, we employed a gelatin zymography assay (**Fig 3D**). Increased MMP9 gelatinase activity was detected in culture supernatants isolated from PC9 cells cultured with MSCs compared to those supernatants isolated from MSC or PC9 cells grown as single cultures (**Fig 3D and 3E**). In contrast, MMP2 gelatinase activity exhibited high basal levels in supernatants from MSCs alone and was not significantly elevated in supernatants isolated from the MSC-PC9 cancer cell co-cultures (**Fig 3D–3F**). Similar results were observed upon co-culture of HCC827 lung cancer cells with MSCs resulting in enhanced MMP9 but not MMP2 gelatinase activity (**S4E–S4G Fig**).

## ABL kinases are activated in MSC-primed lung cancer cells and are required for MSC-induced MMP9 secretion and gelatinase activity

The ABL kinases, ABL1 and ABL2, promote invasion and metastasis of breast and lung cancer cells [19–21]. ABL kinases are hyper-active in solid tumors in response to diverse signals

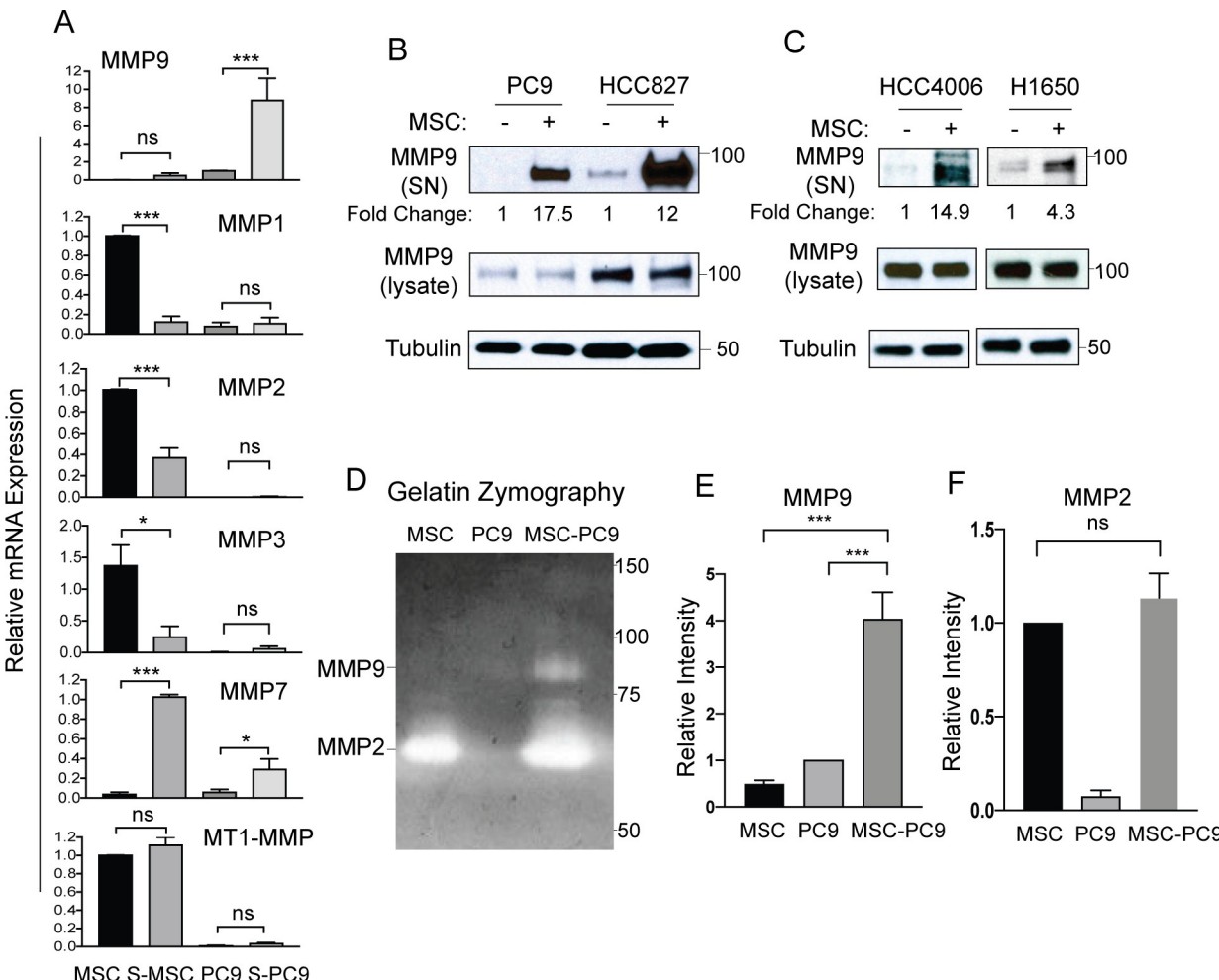

**Fig 3. MSCs enhance expression and secretion of MMP9 in lung cancer cells.** (A) PC9 cells were cultured with or without MSCs followed by FACS to separate sorted PC9 (S-PC9) and sorted MSC (S-MSC) cells from co-culture. RT-PCR analysis was performed for expression of indicated *MMP* mRNAs. Statistical analysis was performed using One-way ANOVA followed by Tukey's multiple comparison post hoc testing (*p<0.05, **p<0.01, ***p<0.001, ns = not significant). Error bars represent ± SEM (n = 3). (B) PC9 or HCC827 lung cancer cells were cultured with or without MSCs for 3 days, after which media was changed to serum-free conditions for an additional 24h. Culture supernatant (SN) and total cell lysates were analyzed by western blotting with indicated antibodies. Supernatant MMP9 was normalized to total MMP9 in lysate and presented as fold change. (C) HCC4006 or H1650 lung cancer cells were cultured with or without MSCs and analyzed as in (B). (D) Gelatin-zymography assay measuring MMP9 activity of culture supernatants from MSC or PC9 single culture or MSC+PC9 co-culture conditions. MMP9 (93kd) and MMP2 (63kd) proteins are indicated. (E-F) Quantification of MMP9 (E) and MMP2 (F) activities by gelatin zymography assay. Statistical analysis was performed using One-way ANOVA followed by Tukey's multiple comparison post hoc testing (***p<0.001; ns = not significant). Error bars represent ± SEM (n = 4).

including activated receptor tyrosine kinases (RTKs), chemokine receptors, adhesion receptors, as well as metabolic and oxidative stress [29]. ABL kinases interact with cell surface receptors including RTKs and adhesion proteins [21, 30–34]. We previously reported that ABL kinases interact with and regulate the subcellular localization and function of membrane-bound MT1-MMP (MMP14) in breast cancer cells [35], however ABL activation in tumors in response to MSC-mediated intercellular signals has not been reported. To determine whether ABL kinases are activated in lung cancer cells following MSC co-culture and whether functional ABL kinases are implicated in MSC-dependent regulation of MMP expression and function, we measured ABL kinase activation in lung cancer cells grown with and without MSCs. PC9 or HCC827 lung cancer cells were co-cultured with or without MSCs followed by FACS

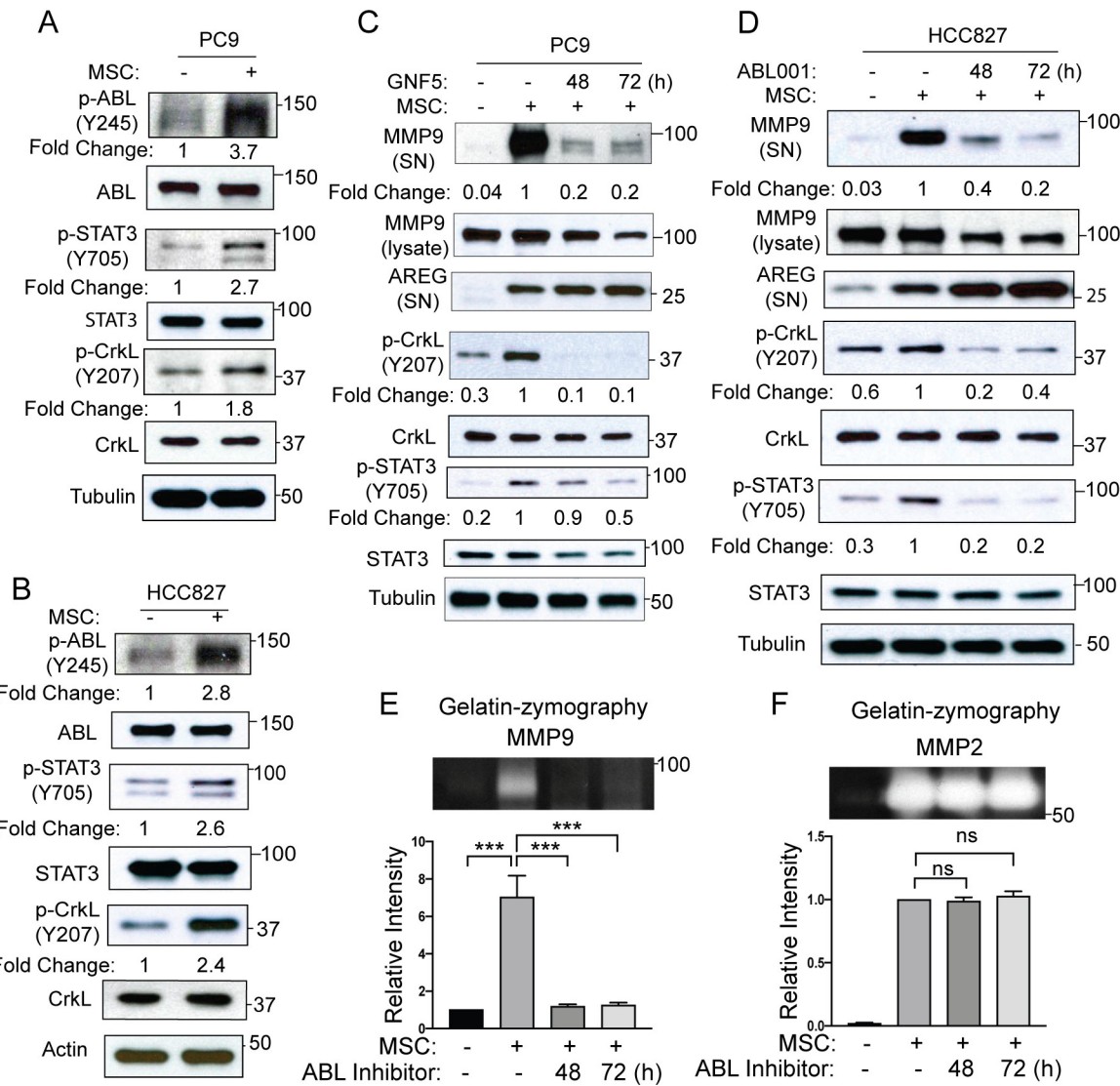

**Fig 4. MSCs induce ABL activation in lung cancer cells which is required for MMP9 secretion and activity.** Immunoblot analysis of PC9 (A) or HCC827 (B) cells cultured with or without MSCs followed by FACS sorting. ABL kinase activity was measured by tyrosine phosphorylation of ABL (Y245), and phosphorylation of ABL downstream targets, STAT3 (Y705) and CrkL (Y207). Phosphorylated protein was normalized to its corresponding total protein and presented as fold change under phospho-protein blots. (C-D) PC9 (C) or HCC827 (D) cells were cultured with or without MSCs and in the presence or absence of ABL kinase inhibitor GNF5 (5 μM) or ABL001 (10 μM) for 48 or 72h. Culture supernatant (SN) was analyzed for MMP9 and Amphiregulin (AREG) proteins. Total cell lysates were analyzed for p-CrkL and p-STAT3 to confirm Abl kinase activity inhibition by GNF5 or ABL001. Supernatant MMP9 was normalized to total MMP9 in lysates and presented as fold change. Phosphorylated protein was normalized to its corresponding total protein and presented as fold change under phospho-protein blots. (E-F) Culture supernatants collected from PC9 cells cultured with or without MSCs in the presence or absence of the ABL kinase inhibitor GNF5 were analyzed for MMP9 and MMP2 activity on gelatin zymography assay. A representative zymographic band is shown (top), and quantification of corresponding proteins (bottom) was performed with Fiji software. Statistical analysis was performed using One-way ANOVA followed by Tukey's multiple comparison post hoc testing. (***p<0.001; ns = not significant). Error bars represent ± SEM (n = 3).

sorting. Immunoblot analysis of sorted lung cancer cells from MSC co-cultures revealed activation of ABL kinases as measured by ABL tyrosine (Y245) phosphorylation and phosphorylation of ABL downstream targets STAT3 (Y705) and CrkL (Y207) (**Fig 4A and 4B**). Thus, MSCs promote ABL activation and downstream signaling in lung cancer cells.

To determine whether ABL kinase activity is required for MSC-induced MMP9 secretion by lung cancer cells, PC9 or HCC827 cells were cultured with or without MSCs in the absence or presence of the ABL-specific allosteric inhibitors GNF5 or ABL001 (also known as Asciminib). These small molecule inhibitors bind to the myristate-binding site in the ABL tyrosine kinase domain independently of the ATP-binding site [36]. Treatment with GNF5 or ABL001 greatly reduced MMP9 secretion in co-cultures of MSCs with either PC9 or HCC827 (**Fig 4C and 4D**; **S5A and S5B Fig**) lung cancer cells. Gelatin zymography assays showed decreased MMP9 gelatinase activity but not MMP2 activity in the presence of ABL allosteric inhibitors in co-cultures of MSCs with either PC9 (**Fig 4E and 4F**) or HCC827 (**S5C and S5D Fig**). Notably, the effect of ABL kinase inhibition on MMP9 secretion appears to be selective as secretion of amphiregulin (AREG), which is markedly enhanced by co-culture of MSCs with lung cancer cells, was not decreased by treatment with either GNF5 or ABL001 (**Fig 4C and 4D**). These findings reveal MSC-induced activation of ABL kinases in lung cancer cells and show that ABL kinase activity is required for MMP9 secretion induced under co-culture conditions.

## Expression of ABL kinases in lung cancer cells is required for MSC-induced MMP9 expression and secretion

To assess whether decreased MMP9 expression and function by ABL allosteric inhibitors in co-cultures of MSCs with cancer cells is mediated by inactivation of ABL kinase activity specifically in the lung cancer cells, PC9 and HCC827 cells were transduced with lentiviruses encoding scramble control shRNA (SCR) or ABL1+ABL2 shRNAs (AA), and these cells were cultured with or without MSCs followed by FACS sorting for analysis. Expression of *MMP9* was greatly increased in PC9 scrambled control (SCR) cells co-cultured with MSCs versus PC9 cells grown in the absence of MSCs, and knockdown of ABL kinases in PC9 (AA) lung cancer cells markedly decreased the MSC-mediated induction of *MMP9* mRNA expression (**Fig 5A**). Further, depletion of ABL kinases in lung cancer cells decreased the MSC-mediated expression of secreted MMP9 protein in PC9 (**Fig 5B**) and HCC827 (**S6A Fig**) lung cancer cells compared to the corresponding control cells. These results were further supported by analysis of MMP9 protein production using an angiogenesis protein array, which showed decreased MMP9 protein levels in culture supernatants from ABL-knockdown PC9 cells cultured in the presence of MSCs versus PC9 control cells co-cultured with MSCs (**S6B Fig**). Depletion of ABL1 and ABL2 protein kinases in PC9-AA and HCC827-AA lung cancer cells impaired the increase of MMP9 but not MMP2 gelatinase activity induced by MSCs (**Fig 5C and 5D**; **S6C and S6D Fig**). These data demonstrate that ABL1 and ABL2 kinases expressed in lung cancer cells are required for MSC-induced MMP9 expression and secretion.

## High expression of *MMP9* is linked to low survival rate in lung adenocarcinoma patients

MMP9 has been shown to promote metastatic niche formation and metastatic colonization in mouse models [37]. Increased *MMP9* expression correlates with aggressive breast cancer with higher incidence of relapse and metastasis [38, 39]. A previous study also identified a correlative relationship between NSCLC metastasis with tumor MMP9 expression [40]. Thus, we performed analysis of lung adenocarcinoma patient microarray data for overall survival (OS) and progression-free survival (PFS) (**Fig 6**). Analysis of 719 patients for OS and 461 patients for PFS demonstrated that high expression of *MMP9* significantly correlated with low survival. These results suggest a role for MMP9 in lung adenocarcinoma progression and metastasis associated with poor prognosis.

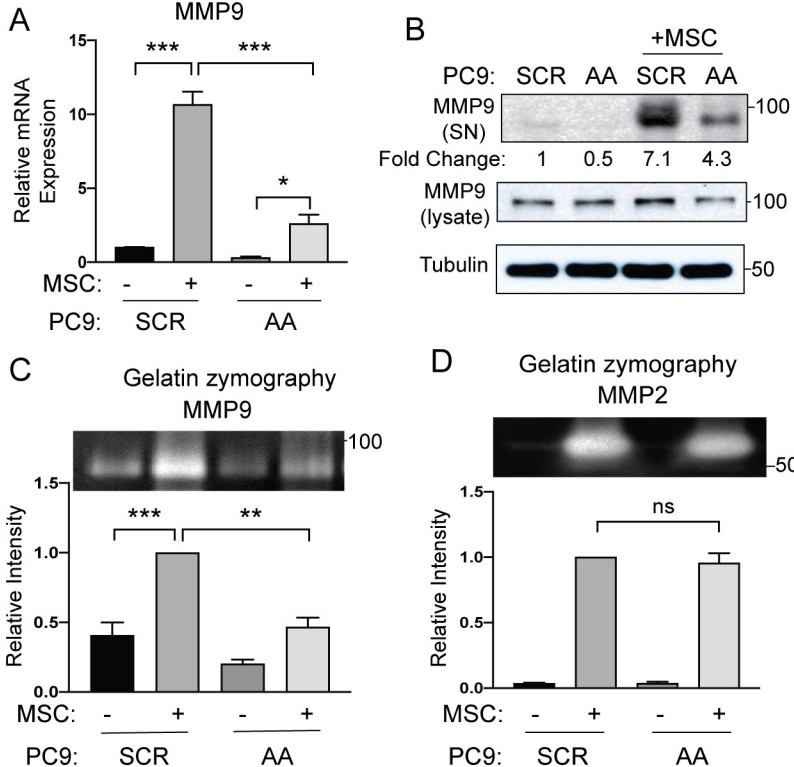

**Fig 5. Depletion of ABL kinases reduces expression and secretion of MMP9.** PC9 cells were transduced with lentiviruses encoding either scramble control shRNA (SCR) or shRNAs specific for ABL1 and ABL2 (AA). Cells were then co-cultured with or without MSCs followed by FACS sorting to separate PC9-SCR or PC9-AA cells from MSCs. (A) RT-PCR analysis of PC9-SCR, PC9-AA, sorted PC9-SCR and sorted PC9-AA cells for *MMP9* mRNA expression. Statistical analysis was performed using One-way ANOVA followed by Tukey's multiple comparison post hoc testing (***$p < 0.001$; *$p < 0.05$). Error bars represent ± SEM (n = 4). (B) Culture supernatants (SN) from PC9-SCR or PC9-AA cells grown alone or with MSCs under co-culture conditions were analyzed by Western blotting for MMP9 protein and total cell lysates were blotted with indicated antibodies. MMP9 protein in supernatants was normalized to total MMP9 in lysates and presented as fold change. (C-D) Culture supernatants of PC9-SCR or PC9-AA with or without MSC co-culture were analyzed for MMP9 (C) and MMP2 (D) using gelatin-zymography. A representative zymographic band is shown (top) with corresponding band quantification (bottom). Statistical analysis was performed using One-way ANOVA followed by Tukey's multiple comparison post hoc testing (***$p < 0.001$, **$p < 0.01$, ns = not significant). Error bars represent ± SEM (n = 3).

## ABL kinases are required for MSC-induced NSCLC metastasis

We reported that enhanced expression of ABL kinases and ABL-regulated targets in lung cancer cells is associated with decreased survival for lung adenocarcinoma patients [19, 21]. To evaluate ABL kinase involvement in MSC-induced NSCLC metastasis, PC9 and HCC827 lung cancer cells were transduced with lentiviruses encoding either scrambled control shRNA (SCR) or shRNAs specific for ABL1 and ABL2 (AA) (**Fig 7**). Efficient knockdown of ABL1 and ABL2 proteins and decreased phosphorylation of the ABL substrate p-CrkL was observed in both PC9 and HCC827 lung cancer cells (**Fig 7F and 7G**). Control (SCR) and ABL1+ABL2 knockdown (AA) lung cancer cells were co-cultured with MSCs and then implanted in mice by intracardiac injection. Depletion of ABL kinases in PC9 and HCC827 lung cancer cells impaired MSC-induced metastasis (**Fig 7A–7D**). Quantification of total body bioluminescence imaging (BLI) counts showed a profound decrease in metastasis in mice injected with MSC-primed ABL-knockdown lung cancer cells compared to mice injected with MSC-primed control cells (**Fig 7B–7E**). Further, mice injected with ABL-depleted PC9 lung cancer cells pre-

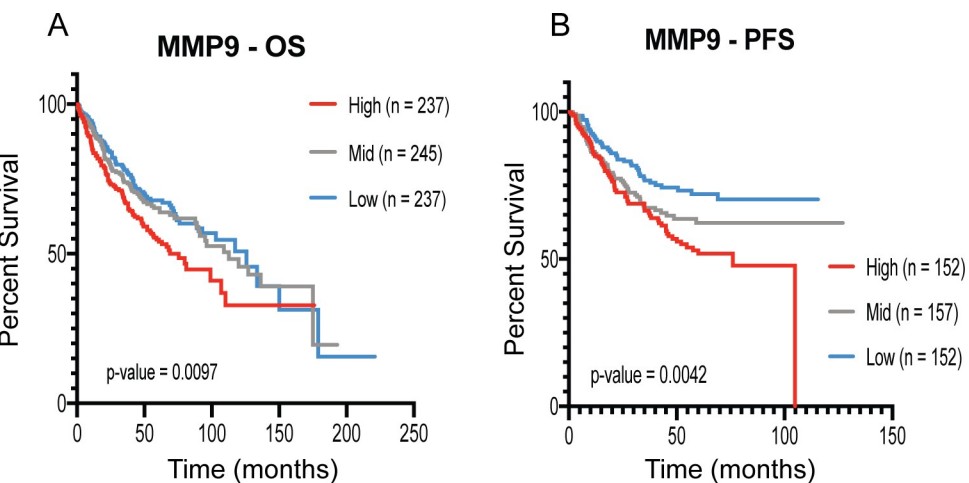

**Fig 6. High *MMP9* expression correlates with poor survival in lung adenocarcinoma patients.** Analysis of lung adenocarcinoma patient microarray data for *MMP9* expression was performed using the KM plot analysis tool (kmplot.com). Patient groups were divided into tertiles for high, medium and low *MMP9* expression. (A) Overall survival (OS) (n = 719 total patients across all cohorts for OS) and (B) Progression-free survival (PFS) (n = 461 total patients across all cohorts for PFS) for lung adenocarcinoma patients. Statistical analysis was performed by Log-Rank Mantel-Cox testing.

cultured with MSCs exhibited increased survival over mice injected MSC pre-cultured control (SCR) PC9 lung cancer cells (**Fig 7C**). These results support a role for ABL kinases in MSC-induced NSCLC metastasis.

## Discussion

MSCs migrate to primary tumors where the tumor-activated MSCs may induce tumor growth, survival and angiogenesis, and subsequently these MSCs can promote metastatic dissemination and colonization of distal sites [6]. MSCs engage in bidirectional signaling with metastatic tumors through paracrine signaling or direct inter-cellular interactions through cell surface receptors [41]. Accumulating data support a role for MSCs in lung tumor progression and metastasis [42]. However, the signaling networks activated in lung cancer cells primed by intercellular signals from MSCs are just beginning to be elucidated.

To assess the role of MSCs in NSCLC metastasis, we employed a panel of EGFR-mutant human lung adenocarcinoma cells primed by pre-incubation with or without bone marrow-derived MSCs. We found a marked increase in metastasis following intracardiac injection of EGFR-mutant PC9, HCC827, and HCC4006 lung cancer cells that were primed by co-culture with MSCs versus those grown in the absence of MSC-induced priming. The increased metastasis induced by MSCs was not due to increased growth of lung cancer cells co-cultured with MSCs as assayed in vitro. Our findings support a pro-metastatic function by MSC-mediated intercellular signaling with lung adenocarcinoma cells.

Here we identified MMP9 as a target of MSC-induced priming in lung adenocarcinoma cells. Following co-culture with MSCs and FACS sorting, we detected a profound and selective induction of *MMP9* mRNA expression in lung cancer cells among the cohort of *MMP* transcripts evaluated. The MSC-induced increase in MMP9 transcription resulted in increased secretion of MMP9 protein and a corresponding increase in MMP9 gelatinase activity among a panel of lung adenocarcinoma cells. In contrast, MMP2 gelatinase activity was not altered in lung cancer cells primed by MSCs under co-culture conditions. MMP9 has been shown to modulate distinct steps in the metastatic cascade including tumor invasion, metastatic niche

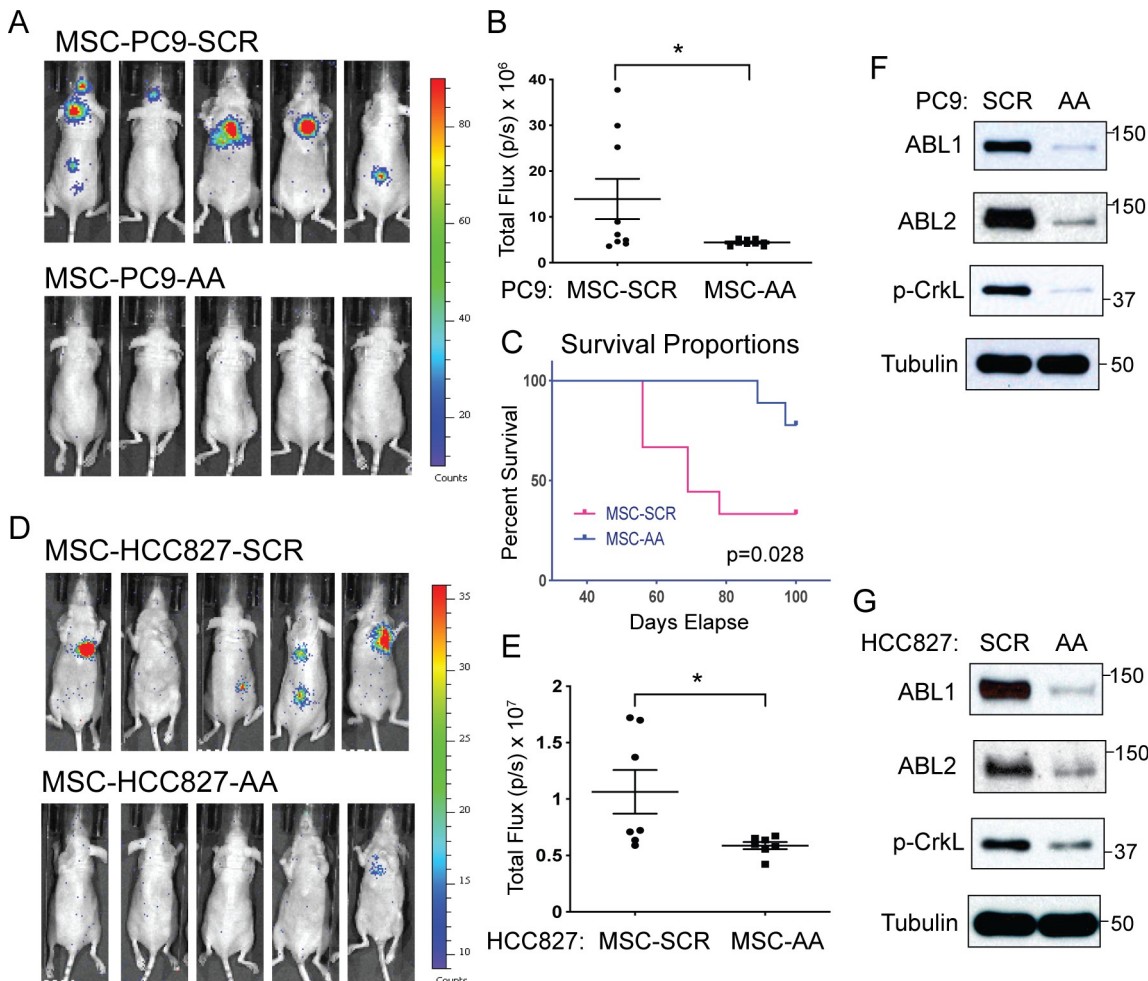

**Fig 7. Depletion of ABL kinases in lung cancer cells abrogates MSC-induced increase of metastatic activity.** PC9 or HCC827 cells labeled with luciferase were transduced with viruses carrying either scramble control shRNA (SCR) or shRNAs specific for ABL1 and ABL2 (AA). Cells were then cultured with MSCs (1:1 ratio) for 3 days. Equal number of cells were intracardially-injected into nude mice ($4x10^5$/mouse for PC9 or $1x10^5$ /mouse for HCC827). (A-B) Representative BLI images at day 21 post injection are shown (A) and BLI total flux (p/s) counts were plotted (B) (n = 9 for each group) for mice injected with MSC+PC9-SCR or MSC+PC9-AA cells. (C) Percent survival for mice injected with MSC+PC9-SCR vs. MSC+PC9-AA cells over 100 days. Survival statistical analysis was calculated using Log rank (Mantel Cox) test (n = 9 each group). (D-E) Representative BLI images at day 24 post injection are shown (D) and BLI total flux (p/s) counts were plotted (E) (n = 7 for each group) for mice injected with MSC+HCC827-SCR or MSC +HCC827-AA cells. Total flux p values were calculated with unpaired two-tailed T test (* p<0.05). (F-G) Total lysates were probed with indicated antibodies to assess knockdown of ABL1 and ABL2 in PC9 (F) and HCC827 (G) cells transduced with lentiviruses carrying either control (SCR) or shRNAs specific for ABL1 and ABL2 (AA).

formation, angiogenesis, and metastatic colonization of secondary sites [28]. MMP9 is expressed in CD11b+Gr1+ myeloid cells localized in the lungs of a mammary tumor virus (MMTV) promoter-driven polyoma middle T antigen (PyMT) mouse model of breast cancer, and MMP9 activity plays a role in the initiation and maintenance of the metastatic niche in the lungs of tumor-bearing mice [37]. In contrast, others reported that MMP9 is produced by human breast cancer cells and is required for pulmonary metastasis in a mouse orthotopic model of basal-like breast cancer [43]. These data are consistent with our findings showing increased production of MMP9 in metastatic lung cancer cells primed by co-culture with MSCs. Future studies using single-cell RNA-sequencing approaches will be needed to assess whether increased expression of MMP9 in either cancer cells or distinct cell types in the tumor

microenvironment may be linked to poor outcomes and to dissect the contribution of activated MMP9 to NSCLC metastatic dissemination and colonization of multiple organ sites including the brain which is a common site for NSCLC metastasis.

Several studies have found that the presence of high stroma in NSCLC tumors correlated with increased risk of relapse and poor prognosis [44, 45]. Our analysis of a cohort of lung adenocarcinoma patients demonstrated that high expression of *MMP9* significantly correlated with decreased overall survival and progression-free survival. These data suggest that MMP9 could be a potential therapeutic target for treating metastatic lung adenocarcinoma patients. However, effective therapies targeting MMP9 have failed to extend survival for patients with breast cancer and other tumors [46, 47]. Thus, therapies are needed that block the function of MMP9 and additional targets in cancer cells primed by MSCs.

Unexpectedly, we found that ABL tyrosine kinases are activated in lung cancer cells primed by MSCs and that inactivation of ABL kinases impairs MSC-induced MMP9 secretion and gelatinase activity. Tumor-intrinsic activation of ABL kinases has been detected in multiple solid tumors [29]. Here we show that ABL kinases are activated in lung cancer cells in response to MSC-mediated intercellular signals. Moreover, we show that ABL expression and kinase activity are required for MSC-induced MMP9 secretion by lung cancer cells. Both ABL knockdown and treatment of lung cancer cells with ABL-specific allosteric inhibitors markedly reduced MMP9 secretion and gelatinase activity. The inhibitory effect of ABL kinase inhibition on MMP9 secretion is selective because loss of ABL function did not impair secretion of other MMPs or the EGFR ligand amphiregulin (AREG), which is induced by MSC-priming in lung cancer cells.

These findings reveal for the first time MSC-induced activation of ABL kinases in lung cancer cells. Because ABL allosteric inhibitors are currently in clinical trials for the treatment of therapy-resistant leukemia patients [48], these inhibitors could be employed to impair MMP9 secretion and function in metastatic lung adenocarcinoma cells primed by MSCs. Importantly, we found that depletion of ABL kinases in lung adenocarcinoma cells impaired MSC-induced metastasis and increased survival in pre-clinical mouse models. Together these data revealed an actionable ABL-MMP9 signaling axis to inhibit lung adenocarcinoma metastasis induced by priming of lung cancer cells by MSCs in the tumor micro-environment.

## Supporting information

**S1 Fig. Growth of PC9 lung cancer cells is not altered by MSC priming.** PC9 cells labeled with pFU-luciferase tomato were co-cultured with Cell-Tracker violet-labeled MSCs at 1:1 ratio. (A) Representative flow cytometry cell sorting to separate PC9 cells from MSCs after 3 days of co-culture. (B) Comparison of MSC and PC9 cell numbers grown under single culture (S) versus co-culture (Co) conditions and separated by FACS (n = 3).
(TIF)

**S2 Fig. MSCs enhance metastasis of HCC4006 lung cancer cells under co-culture conditions.** (A) HCC4006 cells labeled with luciferase were cultured with or without MSCs (1:1 ratio). Equal number of cells were used for intracardiac injection and BLI imaging was performed at day 28 post-injection. (B) Total flux (p/s) counts was plotted (n = 5). Statistical p value was calculated using unpaired one-tailed t test.
(TIF)

**S3 Fig. MSCs promote EMT in lung cancer cells.** HCC827, HCC4006, H1650 and PC9 lung cancer cells were cultured with or without MSCs followed by FACS sorting. RT-PCR for indicated EMT markers was performed with lung cancer cells isolated from single culture

compared to lung cancer cells sorted from co-culture with MSCs. Statistical analysis was performed using unpaired two-tail t test. $^*$p<0.05, $^{**}$p<0.01, $^{***}$p<0.001; ns = not significant. All assays were done in triplicate.
(TIF)

**S4 Fig. MSCs promote *MMP9* expression and increase MMP9 gelatinase activity in NSCLC cells.** (A) RT-PCR for *MMP9* mRNA expression in PC9, HCC827, HCC4006, and H1650 lung cancer cells cultured with or without MSCs followed by FACS sorting. Statistical analysis was performed using unpaired two-tail t test. $^*$p<0.05, $^{**}$p<0.01, $^{***}$p<0.001. All assays were done in triplicate. (B-C) RT-PCR for *MMP9* and *MMP7* mRNA expression in PC9 (B) and H1650 (C) cancer cells cultured with or without MSCs followed by FACS sorting. Statistical analysis was performed using One-way ANOVA followed by Tukey's multiple comparison post hoc analysis ($^{***}$p<0.001; $^{**}$p<0.01). (D) PC9 cells were cultured with or without MSCs in the presence or absence of ABL kinase inhibitor GNF5 (5 μM) for 48 or 72h. Culture supernatants (SN) from MSC alone or PC9 co-cultured with or without MSC in the presence or absence of GNF5 were analyzed for MMP9 and MMP7 proteins. Total lysates were blotted with MMP9, MMP7 and tubulin. (E) Culture supernatants from MSCs, HCC827 single culture, or MSC+-HCC827 co-culture were analyzed for MMP9 activity by gelatin-zymography assay. MMP9 and MMP2 gelatin digestion bands were indicated. (F-G) Quantification of MMP9 (F) and MMP2 (G) was carried out by Fiji software. Statistical analysis was performed using One-way ANOVA followed by Tukey's multiple comparison post hoc testing. ($^{***}$p<0.001; ns = not significant). Error bars represent ± SEM (n = 3).
(TIF)

**S5 Fig. Allosteric inhibition of ABL kinase activity reduces MMP9 secretion and function.** (A) HCC827 cells were cultured with or without MSCs and in the absence or presence of ABL allosteric inhibitor GNF5 (10 μM) for 72 h. Culture supernatants (SN) were analyzed for MMP9 protein and normalized to tubulin presented as fold change. (B) PC9 cells were cultured with or without MSCs and in the presence or absence of ABL allosteric inhibitor ABL001 (5 μM) for 48 and 72 h. Culture supernatants (SN) were analyzed for MMP9 and AREG proteins. MMP9 proteins in supernatant were normalized to MMP9 proteins in the lysate and presented as fold change. Total cell lysates were also analyzed with the indicated antibodies. (C-D) Culture supernatants collected from HCC827 cells cultured with or without MSCs in the presence or absence of ABL allosteric inhibitors ABL001 were analyzed for MMP9 activity on gelatin zymography. A representative zymographic band is shown (top), and quantifications of corresponding bands (bottom) was done by Fiji software. Statistical analysis was performed using One-way ANOVA followed by Tukey's multiple comparison post hoc testing ($^{**}$p<0.01, $^*$p<0.05, ns = not significant). Error bars represent ± SEM (n = 2).
(TIF)

**S6 Fig. Knockdown of ABL kinases reduces MMP9 secretion and function.** (A) HCC827 lung cancer cells were transduced with either scramble control shRNA (SCR) or shRNAs specific for ABL1 and ABL2 (AA). Cells were then cultured with or without MSCs. Culture supernatants (SN) were analyzed for MMP9 protein and normalized to MMP9 in lysates and expressed as fold change. (B) PC9-SCR and PC9-AA cells were cultured with or without MSCs, and culture supernatants were analyzed for MMP9 protein using the Angiogenesis Array performed in duplicates. Statistical analysis was performed using One-way ANOVA followed by Tukey's multiple comparison post hoc testing ($^{**}$p<0.01; $^{***}$p<0.001). (C-D) Cell culture supernatants from HCC827 cells transduced with shRNA control (SCR) or shRNAs-specific against ABL1+ABL2 (AA) were cultured with or without MSCs and were then

analyzed for MMP9 (C) and MMP2 (D) enzymatic activity using gelatin-zymography. Representative zymographic band are shown (top), and quantification of corresponding bands (bottom) was carried out by Fiji software. Statistical analysis was performed using One-way ANOVA followed by Tukey's multiple comparison post hoc testing (***$p < 0.001$, **$p < 0.01$, ns = not significant). Error bars represent ± SEM (n = 2).
(TIF)

**S1 Table. Primers used for RT-PCR.**
(XLSX)

**S2 Table. Antibodies for Western blots analysis.**
(XLSX)

**S1 File.**
(PDF)

## Acknowledgments

We thank the Duke University Flow Cytometry Shared Resource for assistance with cell sorting.

## Author Contributions

**Conceptualization:** Jing Jin Gu, Ann Marie Pendergast.

**Data curation:** Jing Jin Gu.

**Formal analysis:** Jing Jin Gu, Jacob Hoj, Ann Marie Pendergast.

**Funding acquisition:** Ann Marie Pendergast.

**Investigation:** Jing Jin Gu, Clay Rouse, Ann Marie Pendergast.

**Methodology:** Jing Jin Gu, Clay Rouse, Ann Marie Pendergast.

**Project administration:** Ann Marie Pendergast.

**Resources:** Ann Marie Pendergast.

**Software:** Jacob Hoj.

**Supervision:** Ann Marie Pendergast.

**Writing – original draft:** Jing Jin Gu, Ann Marie Pendergast.

**Writing – review & editing:** Jing Jin Gu, Jacob Hoj, Ann Marie Pendergast.

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
