## [Decision Letter · Decision Letter 0]

25 Aug 2020

PONE-D-20-19649

Mesenchymal stem cells promote metastasis through activation of an ABL-MMP9 signaling axis in lung cancer cells

PLOS ONE

Dear Dr. Pendergast,

Thank you for submitting your manuscript to PLOS ONE. After careful consideration, we feel that it has merit but does not fully meet PLOS ONE’s publication criteria as it currently stands. Therefore, we invite you to submit a revised version of the manuscript that addresses the points raised during the review process.

We look forward to receiving your revised manuscript.

Kind regards,

Srikumar Chellappan

Academic Editor

PLOS ONE

Additional Editor Comments:

The manuscript has been reviewed by two experts in the field. They are enthusiastic about the study, but have suggested certain minor modifications. Please submit a revised version that addresses these issues; the manuscript will be send back to the reviewers for their comments. I look forward to seeing the revised manuscript.

2. Please provide additional information about each of the cell lines used in this work, including any quality control testing procedures (authentication, characterisation, and mycoplasma testing).

For more information, please see http://journals.plos.org/plosone/s/submission-guidelines#loc-cell-lines

3. Please provide additional information about the HEK293T cells used in this work, including the source any quality control testing procedures (authentication, characterisation, and mycoplasma testing).

For more information, please see http://journals.plos.org/plosone/s/submission-guidelines#loc-cell-lines

4. To comply with PLOS ONE submissions requirements, please provide the method of euthanasia in the Methods section of your manuscript.

Reviewers' comments:

Reviewer's Responses to Questions

**Comments to the Author**

1. Is the manuscript technically sound, and do the data support the conclusions?

Reviewer #1: Yes

Reviewer #2: Yes

2. Has the statistical analysis been performed appropriately and rigorously? 

Reviewer #1: Yes

Reviewer #2: Yes

3. Have the authors made all data underlying the findings in their manuscript fully available?

Reviewer #1: Yes

Reviewer #2: Yes

4. Is the manuscript presented in an intelligible fashion and written in standard English?

Reviewer #1: Yes

Reviewer #2: Yes

5. Review Comments to the Author

Reviewer #1: The article by Gu and colleagues titled “Mesenchymal stem cells promote metastasis through activation of an ABL-MMP9 signaling axis in lung cancer cells” dissects the interplay between MSCs and lung cancer with respect to tumor progression and metastasis using a combination of in vitro and in vivo experiments. Briefly, the key findings of the study are that MSCs promote metastasis in multiple non-small cell lung cancer cells following co-culture, an effect which appears to be independent of changes on growth. Instead, the co-culture of MSCs and NSCLC cells upregulated expression of several pro-metastatic factors, including MMP9. Interestingly, MSC-mediated induction of MMP9 was dependent on ABL kinases. These findings were nicely demonstrated using both molecular (shRNA) and pharmacological (ABL-specific allosteric inhibitors) and validated in two independent NSCLC lines, PC9 and HCC827. The study and experiments have been well designed and are overall presented clearly and effectively. There are only a few minor comments and recommendations to be added to the manuscript prior to publication:

• MMP7 transcript expression was also significantly increased following MSC/NSCLC co-culture (Figure 3A). The authors should clarify why only MMP9 was pursued.

• In Figure 4A and B, pCrkL and pSTAT3 are used as readouts for ABL kinase activity, but only pCrkL is presented in Figure 4C and D to validate the effect of treatment with GNF5 or ABL001. Levels of pSTAT3 should be added to these panels for consistency if possible.

• Line 297 refers to “Affimetrix.” Please check whether this should be “Affymetrix.”

• In contrast to the PC9 cells (Figure 7C), the impact of ABL depletion on survival in HCC827 is not mentioned. Even if not significant, the data for HCC827 could be included for completion, and the authors could discuss potential reasons for the difference.

• Please specify that the bone marrow MSCs from Lonza are human origin in the Materials and Methods.

Reviewer #2: The manuscript entitled “Mesenchymal stem cells promote metastasis through activation of an ABL-MMP9 signaling axis in lung cancer cells” authored by Gu JJ, Hoj J, Rouse C and Pendergast AM is a very interesting report of ABL kinase modulating EGFR-mutant non-small cell lung cancer (NSCLC) invasiveness and metastatic colonization through metalloproteinase (MMP) upregulation. The authors cultivated NSCLC cell line with or without bone marrow-derived mesenchymal stem cells (BM-MSC) for three days, separated cell types by sorting and then performed NSCLC intracardiac injection to assess metastatic dissemination and colonization of tissues. After noticed that NSCLC cell lines primed by BM-MSC have EMT-like phenotype and increased metastatic abilities, they used cellular and molecular biology techniques to identify which MMP might be involved in this phenomenon. MMP 9 mRNA, protein secretion and activity were increased in BM-MSC primed NSCLC cell lines and the authors characterized that ABL kinases were responsible for its upregulation. Supporting the in vivo studies, microarray data showed correlation between high MMP9 expression and poor survival in lung adenocarcinoma patients. Finally, ABL kinases are identified as putative target therapy candidates as ABL kinase inhibitors and depletion could impair MMP9 secretion and activity as well as diminish NCSLC cell metastasis in the in vivo model.

The work is well designed, well performed and I believe it will appeal to PlosOne audience. Only minors suggestions:

1 – BM-MSC tropism to inflammatory and cancer sites is well reported. However the exactly extent that it could happen in pathophysiological conditions it is not known. The work shows a correlation of high MMP9 expression, which is upregulated by stromal BM-MSC, and poor patient outcome. Have authors assessed how many of these patients have significant tumor stroma associated-lung adenocarcinoma?

2 - It is known that cell passages could influence BM-MSC phenotype and cell outcome. Could the authors cite in which passage cells were used at the material and methods?

3 - In many protein blots, it is shown phosphorylated protein isoform but not total protein isoform. It is known that alterations within the total amount of protein available could affect the interpretation of the measurement of protein phosphorylation status. In line with this, the normalization of a phosphorylated protein to its total expression allows the ratio of phosphorylated proteins to be assessed. Do the authors believe that this normalization is not necessary?

Protein normalization to tubulin is represented as fold change above blots. Is it appropriated to show it? If it is not normalized to respective total protein, why show tubulin normalization results as fold change?

4 - It is very important to watch for figure definition. Some legends are too small or without resolution. Mainly graphic axis legends. It must be improved before final version.

5 - Data regarding lung cancer survival rate in lines 56-57 should be referenced.

6. PLOS authors have the option to publish the peer review history of their article (what does this mean?). If published, this will include your full peer review and any attached files.

Reviewer #1: No

Reviewer #2: **Yes: **Pedro Barcellos-de-Souza

---

## [Author Response · Author response to Decision Letter 0]

7 Oct 2020

Per 8/25/2020 PLOS ONE e-mail instructions, a PDF document (Response to Reviewers) has been uploaded as a separate file.

---

## [Editor Report · Decision Letter 1]

15 Oct 2020

Mesenchymal stem cells promote metastasis through activation of an ABL-MMP9 signaling axis in lung cancer cells

PONE-D-20-19649R1

Dear Dr. Pendergast,

We’re pleased to inform you that your manuscript has been judged scientifically suitable for publication and will be formally accepted for publication once it meets all outstanding technical requirements.

Kind regards,

Srikumar Chellappan

Academic Editor

PLOS ONE
---

## [Editor Report · Acceptance letter]

19 Oct 2020

PONE-D-20-19649R1 

Mesenchymal stem cells promote metastasis through activation of an ABL-MMP9 signaling axis in lung cancer cells 

Dear Dr. Pendergast:

I'm pleased to inform you that your manuscript has been deemed suitable for publication in PLOS ONE. Congratulations! Your manuscript is now with our production department. 

Kind regards, 

on behalf of

Dr. Srikumar Chellappan 

Academic Editor

PLOS ONE